# Cascading effects of thermally-induced anemone bleaching on associated anemonefish hormonal stress response and reproduction

Ricardo Beldade[1,2], Agathe Blandin[1], Rory O'Donnell[3] & Suzanne C. Mills [1,4]

Organisms can behaviorally, physiologically, and morphologically adjust to environmental variation via integrative hormonal mechanisms, ultimately allowing animals to cope with environmental change. The stress response to environmental and social changes commonly promotes survival at the expense of reproduction. However, despite climate change impacts on population declines and diversity loss, few studies have attributed hormonal stress responses, or their regulatory effects, to climate change in the wild. Here, we report hormonal and fitness responses of individual wild fish to a recent large-scale sea warming event that caused widespread bleaching on coral reefs. This 14-month monitoring study shows a strong correlation between anemone bleaching (zooxanthellae loss), anemonefish stress response, and reproductive hormones that decreased fecundity by 73%. These findings suggest that hormone stress responses play a crucial role in changes to population demography following climate change and plasticity in hormonal responsiveness may be a key mechanism enabling individual acclimation to climate change.

[1] EPHE PSL Research University, USR 3278 CRIOBE CNRS-UPVD, BP 1013, Moorea 98729, French Polynesia. [2] MARE—Marine and Environmental Sciences Centre, Faculdade de Ciências da Universidade de Lisboa, Campo Grande, Lisboa 1749-016, Portugal. [3] School of Geography and Environmental Sciences, Ulster University, Cromore Rd, Coleraine, Londonderry BT52 1SA, Northern Ireland. [4] Laboratoire d'Excellence "CORAIL", CRIOBE, Moorea, France. Correspondence and requests for materials should be addressed to S.C.M. (email: suzanne.mills@univ-perp.fr)

Among the most consistently adverse consequences of natural environmental and social stresses (e.g., predation, winter, low resource abundance) is disruption of vertebrate reproductive physiology and behavior[1–4]. A study of snowshoe hares, *Lepus americanus*, in their natural environment demonstrates how the regulatory effects of predator-induced chronic stress impair reproductive function[5]. The stress axis, the hypothalamic–pituitary–adrenal axis (hypothalamic–pituitary–interrenal axis in fish), and subsequent release of glucocorticoid hormones (GCs) are an integrative mechanism that mediates multiple systemic responses to environmental challenges on physiology, behavior, and morphology[2, 4, 6]. The stress axis aims at maintaining stability (homeostasis) despite changing conditions, such that recurring elevations of GCs that lead to a chronically elevated GC baseline, trigger physiological effects that promote immediate survival at the expense of reproduction[7, 8]. Due to their multiple regulatory effects on an individual's biology, GCs play a crucial role in enabling vertebrates (e.g., seabirds[9, 10] and marine iguanas[11, 12]) to cope with and respond to climate change in the wild.

Anthropogenic environmental change has rapidly altered habitats and climate on a global scale[13, 14], modified the relationship between vertebrate populations and their environment, and thus has the potential to have a severe impact on wildlife populations[15]. Ecological and evolutionary responses to climate change include changes to distribution ranges[16–19], morphology[20], and phenology[21–23]. However, the integrative hormonal stress response needs to be given more consideration in order to determine whether GC-mediated effects on reproduction provide both proximate and ultimate explanations for changes in animal abundance under climate change and whether the hypothalamic–pituitary–adrenal (HPA) axis can adjust to climate change. Evidence for causal links between climate change, the physiological hormone response, and demography in wild animals exposed to perturbations is rare[9–12].

Here, we report the endocrine and fitness responses of individual coral reef fish to an acute sea warming event that, coupled with the 2015–2016 El Niño-Southern Oscillation (ENSO), caused bleaching and mortality on coral reefs worldwide. The impacts of elevated sea temperatures on corals and organisms that depend on them have received considerable attention[24]. However, other marine invertebrates, such as sea anemones, exhibit a coral-like symbiosis with dinoflagellates in the genus *Symbiodinium* (zooxanthellae), which can be disrupted by elevated sea surface temperatures. Like corals, anemones experiencing thermal stress expel zooxanthellae from their tissues, lose coloration, and appear white[25]. Also like corals, bleached anemones sometimes recover. The potentially cascading effects of anemone bleaching on associated fish communities are only recently gaining attention. Bleaching reduces anemone abundance and size and has concomitant effects on anemonefish[26–30], small fishes that have an obligate mutualism with host anemones for shelter and successful reproduction[31, 32]. However, the proximate mechanisms causing lethal (declines in abundance[26, 27]) and sublethal (declines in individual sizes of pairs[27], behavioral changes in predator risk assessment, and declines in egg production[29]) effects in anemonefish are unknown.

As the 2016 ENSO warming event was forecast, we began monitoring the health of 30 wild anemones and the spawning of their associated anemonefish pairs in October 2015 and continued every 2 days for a total of 14 months, until December 2016. Here, we present data over 14 months on the monthly spawning of 13 wild pairs prior to the bleaching event: before (5 months), during (5 months), and after (4 months) the bleaching event. We also present data on anemonefish hormonal levels from the same 13 anemones, as well as from anemonefish in an additional 39 anemones, prior to and during the bleaching event (but not after the bleaching event). During the warming event, anemonefish in bleached anemones experience a significant increase in plasma cortisol levels, whereas fish in unbleached anemones do not. Furthermore, reproductive hormones, 11-ketotestosterone (11-KT) and 17β-estradiol, are significantly lower in male and female anemonefish, respectively, in bleached anemones. Finally, anemonefish couples associated with bleached hosts show a striking decrease in fecundity (produce fewer viable eggs, spawn less frequently) compared to before and after the bleaching period. Couples associated to unbleached hosts do not experience any significant change in fecundity compared to the pre-bleaching and post-bleaching periods. Individual differences in GC responses to environmental stressors will be the key factor in understanding whether and how populations can cope with climate change.

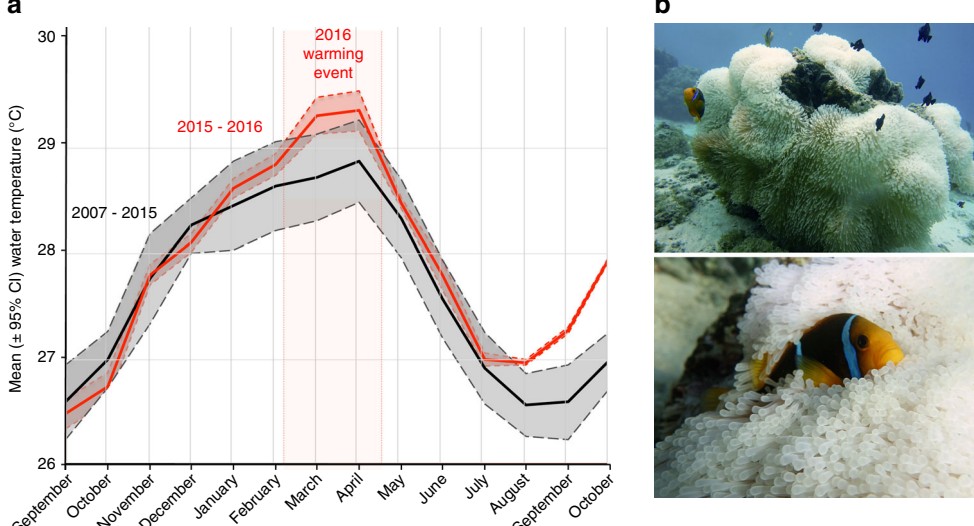

**Fig. 1** Time series of mean water temperature data in Moorea lagoon, French Polynesia, before and during the study. **a** Monthly seawater temperatures over two periods: from 2007 to 2015 (*black*) and from September 2015 to August 2016 (*red*)[61]. Values are mean ±95% confidence intervals. *Vertical red lines* and *shading* represents the 2016 warming anomaly event. **b** Adult anemonefish (*A. chrysopterus*) in bleached anemones (*H. magnifica*)

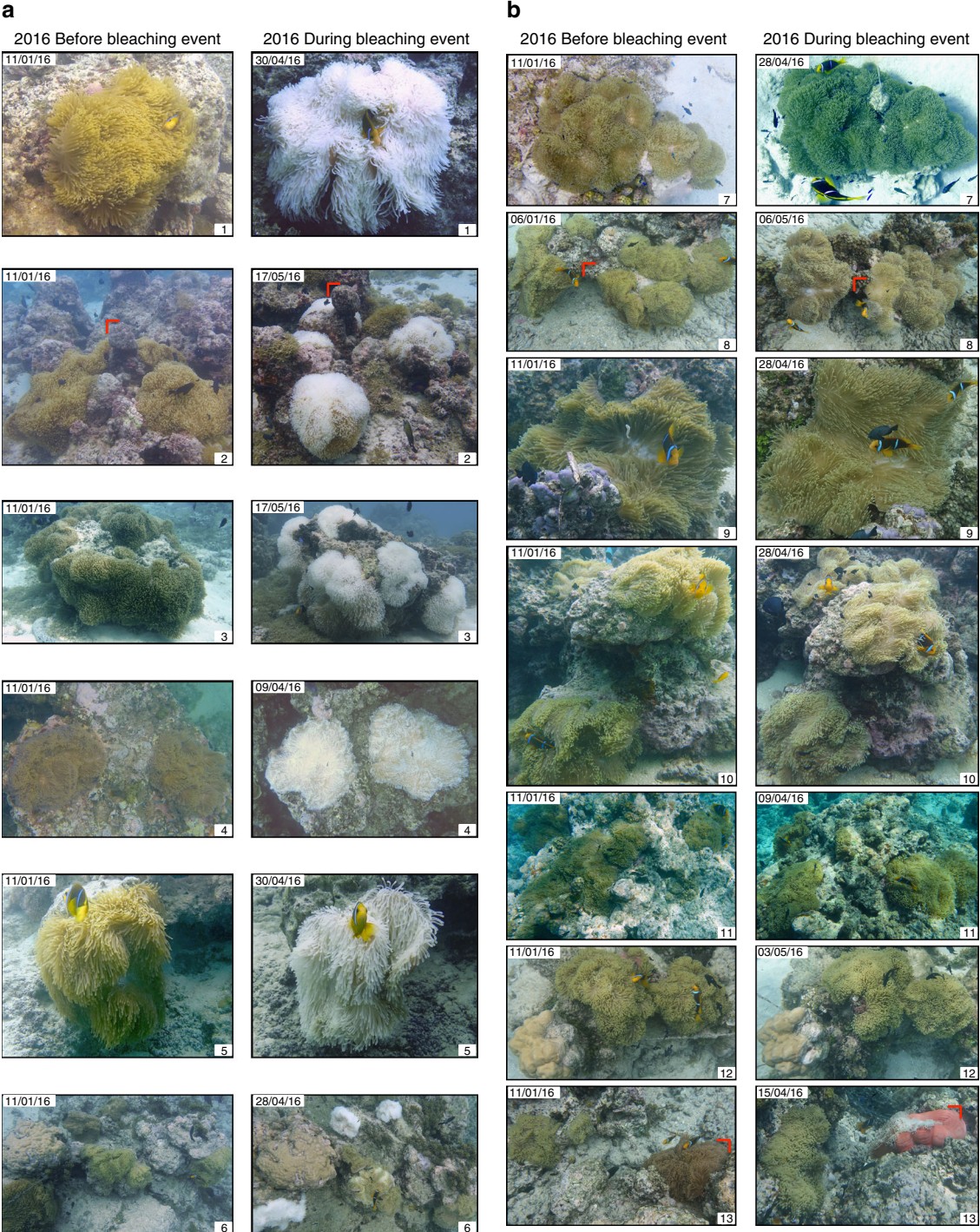

**Fig. 2** Photographs of the 13 anemone *H. magnifica* taken prior to bleaching in January 2016 and during bleaching from April to May 2016. **a** Shows those anemones that bleached and **b** shows those anemones that did not bleach following the warming event in March 2016

## Results

**Thermal anomalies.** During the 2016 ENSO event, water temperature in the lagoon of Moorea, French Polynesia, exceeded historic mean summer temperatures, reaching a mean of 29.3 °C in March 2016 (Fig. 1a). This anomaly corresponded with the onset of *Heteractis magnifica* anemone bleaching (Figs. 1b, 2a), while other individuals remained unbleached (Fig. 2b). Anemone recovery began 4–5 months after the onset of bleaching, and although August–October 2016 temperatures exceeded historic mean winter temperatures, we would not expect continued or new bleaching as the temperatures were well below maximum

summer temperatures. The February/March 2016 thermal event bleached approximately half of the wild anemones hosting orange-fin anemonefish (*Amphiprion chrysopterus*), giving us the unique opportunity to separate the contributions of an environmental perturbation, i.e., anemone bleaching, from those of elevated sea surface temperature, to the physiological response of wild anemonefish.

**Endocrinological impacts in anemonefish.** During the bleaching event, baseline plasma GC levels (cortisol) were significantly elevated in both male and female anemonefish associated with

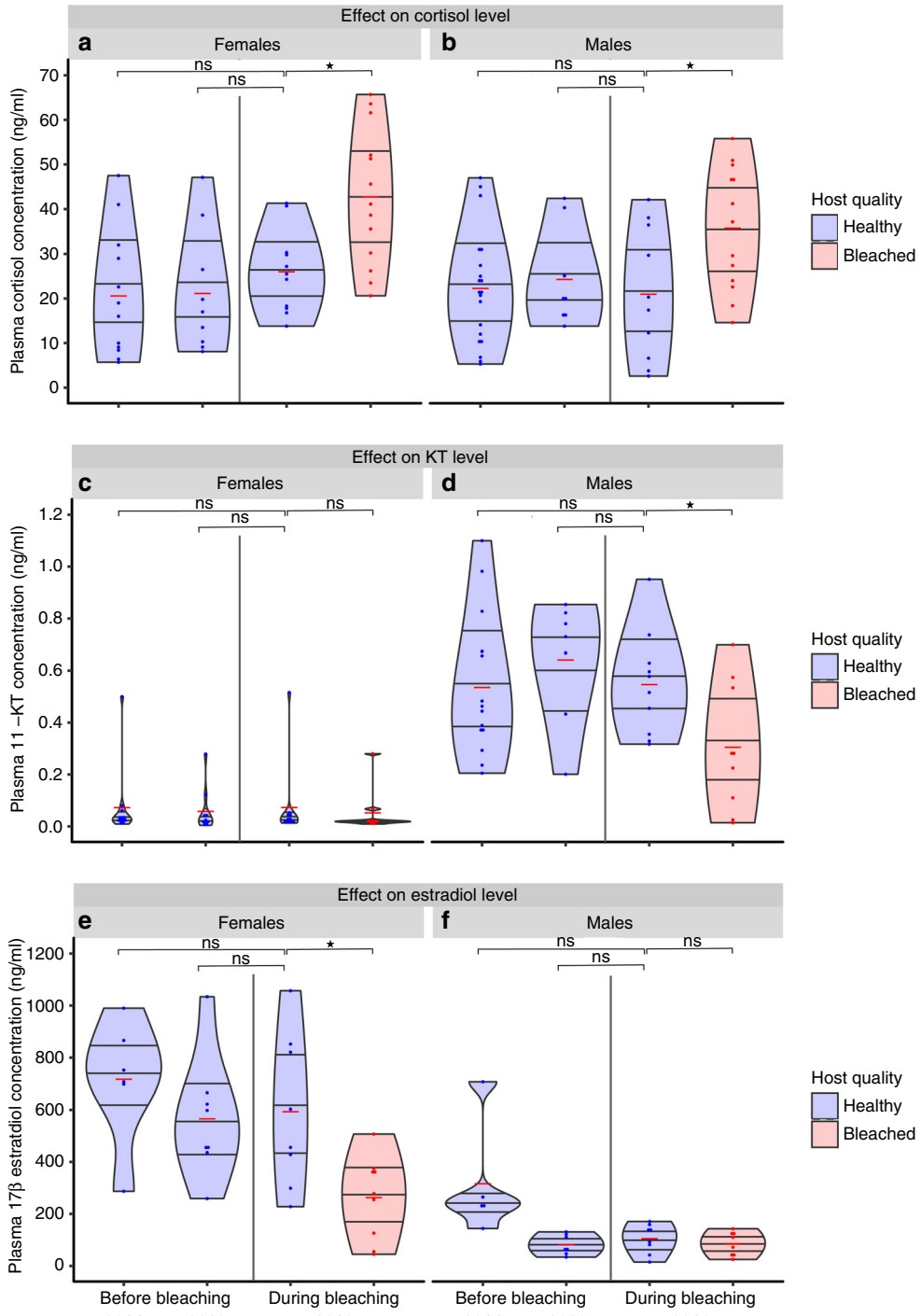

**Fig. 3** Stress and reproductive hormones before and during bleaching events. Effect of exposure to warming-induced anemone bleaching (*red violin plots*) on wild, free-living adult anemonefish plasma hormone concentrations compared with those from adults exposed to unbleached hosts (*blue violin plots*) between three periods: twice before (2014 $n = 12/20$; 2016 $n = 9/8$ for females/males, respectively) and once during the bleaching event (unbleached $n = 11/10$; bleached $n = 13/13$ for females/males, respectively). Hormonal measures were taken from an additional 39 anemonefish pairs that were not monitored for spawning every 2 days. **a** Female cortisol (Linear mixed model (LMM) $F_{3,41} = 7.309$, $P < 0.001$). **b** Male cortisol (LMM $F_{3,47} = 3.464$, $P = 0.023$). **c** Female 11-KT (LMM $F_{3,38} = 0.077$, $P = 0.972$). **d** Male 11-KT (LMM $F_{3,36} = 4.460$, $P = 0.010$). **e** Female 17β-estradiol (LMM $F_{2,27} = 7.840$, $P = 0.001$). **f** Male 17β-estradiol (LMM $F_{2,23} = 2.986$, $P = 0.052$). *Violin plots* show the full distribution of the data with the inner part showing all sample points (*blue dots*), the median (*red line*), and the interquartile range overlaid by the kernel density estimation. Non-significance is denoted by "ns" and statistical significance is denoted by *$P < 0.05$

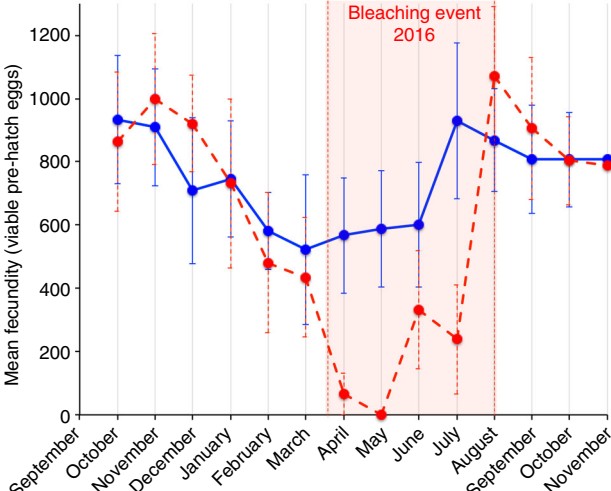

**Fig. 4** Anemonefish fecundity over time including the bleaching event. Effect of exposure to warming-induced anemone bleaching (*vertical red lines* and *shading*) on the monthly fecundity (number of viable pre-hatching eggs) of wild, free-living adult anemonefish over 14 months from October 2015 through November 2016. Pre-hatching fecundity of anemonefish exposed to unbleached ($n = 7$, blue) and bleached ($n = 6$, red) anemones before and during the bleaching event (GLM with repeated measures: month*bleaching $F_{(1,9)} = 8.640$, $P = 0.013$; Supplementary Table 5). Values are mean ± SE

bleached hosts compared to anemonefish in unbleached hosts (Fig. 3a, b; Supplementary Tables 1, 2). Anemonefish baseline cortisol levels from unbleached hosts during the bleaching event were similar to those recorded in anemonefish just prior to and 2 years before the bleaching event (Fig. 3a, b). Levels of 11-KT, the male reproductive hormone, were lower only in male anemonefish associated with a bleached anemone (Fig. 3c, d; Supplementary Tables 1, 3). Conversely, levels of 17β-estradiol, the hormone modulating female reproduction, were lower only in female anemonefish in a bleached host (Fig. 3e, f; Supplementary Tables 1, 4).

**Fitness repercussions and recovery in anemonefish.** As 13 anemonefish pairs that spawned regularly had been monitored for spawning every 2 days since October 2015, and 6 pairs were found exposed to 100% bleached hosts in March/April 2016 (Fig. 2a), it was possible to determine the impacts of bleaching on fecundity over 14 months. The mean monthly number of viable pre-hatch eggs produced by wild anemonefish pairs associated with unbleached anemones showed a non-significant decrease in output in the warmer austral winter months (Fig. 4 and Supplementary Table 5). Prior studies in anemonefish have found both the presence and absence of seasonal periodicity[33, 34]. However, monthly viable pre-hatch egg production plummeted for anemonefish pairs associated with a bleached host, and was significantly lower than that for anemonefish pairs associated with an unbleached host (GLM with repeated measures: during the bleaching event: host bleaching $F_{(1,9)} = 6.397$, $P = 0.028$; Fig. 4 and Supplementary Table 6).

We determined the impacts of bleaching on reproductive function over 14 months: before (October 2015–February 2016), during (March–July 2016), and after (August–November 2016) the bleaching event. Over the 5-month bleaching period, anemonefish associated with bleached hosts spawned less frequently (51% less; Fig. 5a), initially laid 64% fewer eggs (Fig. 5b), experienced 38% higher egg mortality during incubation (Fig. 5c), and ultimately produced 73% fewer viable pre-hatch

eggs (Fig. 5d) compared to before the bleaching period. None of these measures of reproduction differed before, during, or after the bleaching event for anemonefish in unbleached anemones (Fig. 5 and Supplementary Tables 6–8).

The number of viable eggs at hatching of anemonefish in unbleached anemones did not differ before, during, or after the bleaching event (paired *t*-tests, all $P > 0.19$; Supplementary Table 8), whereas clutch size decreased from anemonefish in bleached hosts compared to both before and after (Fig. 5d). Fortunately, all anemones recovered zooxanthellae and regained color after 4–5 months and the number of viable eggs at hatching returned to pre-bleaching levels when the anemones had recovered (before vs. after: $t_5 = 0.613$, $P = 0.567$; Fig. 5d; Supplementary Table 8). This decrease in the number of viable eggs at hatching was due to multiple factors, including a reduction in spawning frequency and the total number of eggs laid, as well as an increase in mortality during incubation.

Spawning frequency significantly decreased during the bleaching event, but only in anemonefish exposed to bleached hosts (Cochran's Q test GLM: $Q_{(1,13)} = 28.0$, $P = 0.009$, Fig. 5a; Supplementary Table 7), but not in those exposed to unbleached hosts (Cochran's Q test GLM: $Q_{(1,13)} = 14.757$, $P = 0.323$, Fig. 5a; Supplementary Table 7). Spawning frequency of anemonefish in unbleached anemones did not differ before, during, or after the bleaching event (Wilcoxon Signed Ranks Test, all $P > 0.078$; Supplementary Table 8), whereas spawning frequency decreased for anemonefish in bleached hosts compared to both before and after (Fig. 5a). Fortunately, spawning frequency returned to pre-bleaching levels when the anemones had recovered (before vs. after: $Z_5 = 1.069$, $P = 0.285$; Fig. 5a; Supplementary Table 8).

Initial clutch size significantly decreased during the bleaching event, but only in anemonefish exposed to bleached hosts (GLM with repeated measures: bleaching $F_{(1,9)} = 6.927$, $P = 0.023$, Fig. 5b; Supplementary Table 6). Initial clutch size of anemonefish in unbleached anemones did not differ before, during, or after the bleaching event (paired *t*-tests, all $P > 0.09$; Supplementary Table 8), whereas clutch size decreased from anemonefish in bleached hosts compared to both before and after (Fig. 5b). Fortunately, initial clutch size returned to pre-bleaching levels when the anemones recovered (before vs. after: $t_5 = 1.040$, $P = 0.346$; Fig. 5b; Supplementary Table 8).

Egg mortality in unbleached anemones did not differ before, during, or after the bleaching event (Wilcoxon Signed Ranks Test, all $P > 0.07$; Supplementary Table 8), whereas egg mortality increased in bleached hosts compared to both before and after (Fig. 5b). Fortunately, egg mortality returned to pre-bleaching levels when the anemones had recovered (before vs. after: $Z_6 = 0.105$, $P = 0.917$; Fig. 5b; Supplementary Table 8).

## Discussion

The thermal disturbance in Moorea thus had a direct effect on anemone endo-symbionts, their zooxanthellae, as well as significant indirect fitness effects on their other symbiont, anemonefish via bleaching. The anemones in our study were spatially interspersed (Supplementary Fig. 1) and experienced the same thermal stress (Supplementary Fig. 2; Supplementary Table 9), therefore their variable bleaching responses may be due, among others, to the different abilities of specific *Symbiodinium* clades to cope with temperature stress, as found for corals[35], as well as micro-scale environmental processes[24]. As such, the differential fitness effects on anemonefish are due to differences in host bleaching and not on the anemonefish themselves, as they all experienced a similar thermal stress and highlight that the impacts of environmental perturbations on one species may be exacerbated for their associated species in symbiotic

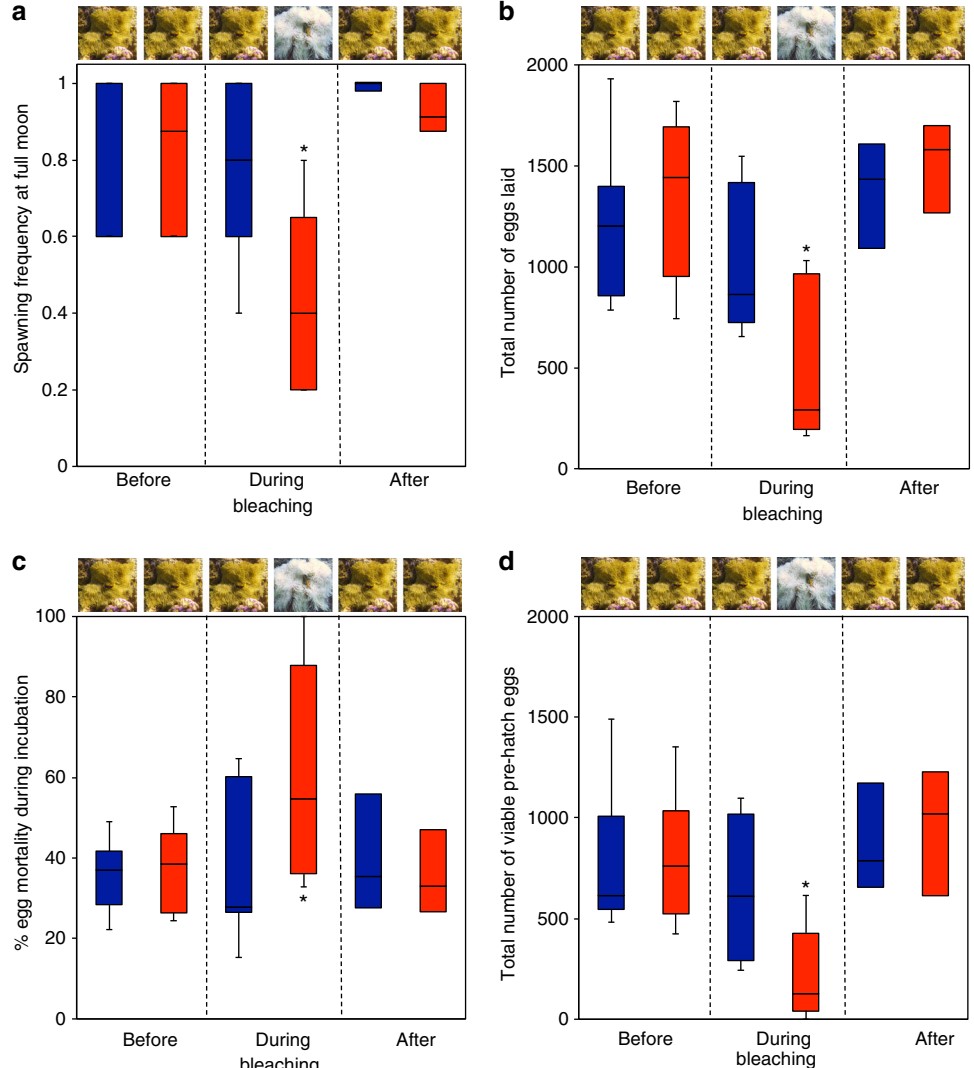

**Fig. 5** Effect of exposure to warming-induced anemone bleaching on free-living adult anemonefish reproductive function during the bleaching event compared to before and after. **a** Spawning frequency (Wilcoxon Signed Ranks Test: before, $Z_5 = 2.003$, $P = 0.045$; after, $Z_5 = 2.207$, $P = 0.027$). **b** Total number of eggs laid per full moon (Paired $t$-tests: before, $t_5 = 4.965$, $P = 0.004$; after, $t_5 = 5.205$, $P = 0.003$). **c** Total proportion of egg mortality (Wilcoxon Signed Ranks Test: before, $Z_5 = 1.992$, $P = 0.046$; after, $Z_5 = 1.992$, $P = 0.046$). **d** Total number of viable pre-hatch eggs (Paired $t$-tests: before, $t_5 = 6.585$, $P = 0.001$; after, $t_5 = 3.897$, $P = 0.011$). Anemonefish before, during and after being exposed to bleached (*red boxplots*) and unbleached (*blue boxplots*) anemones. Values are median, interquartile range, minimum, and maximum. *Asterisk* denotes significance $P < 0.05$

interactions[36]. Why anemonefish should exhibit a stress response to host bleaching is not known, but may be in response to a perceived increased risk of predation either from the shrinking of the anemone, or a reduced neurotoxicity of venom from bleached anemones. Alternatively their increased conspicuousness might lead to increased harassment from heterospecifics[29].

Our results suggest that anemonefish adjust their baseline endocrine phenotype to match current conditions of their host anemone. Elevated baseline GCs are common in response to a wide array of environmental perturbations[37–40] and baseline levels of GCs show a positive relationship with environmental challenges[41, 42] (however see ref. [43]). As such our results of elevated baseline GCs suggest that anemonefish within bleached hosts are chronically stressed. Elevated baseline GCs in anemonefish in response to host bleaching in turn modulate reproduction by lowering levels of reproductive hormones. Such effects, linking cortisol to sex hormones and then linking cortisol to reproduction, have been documented in other fish species in laboratory studies[44], but no studies have documented the pathway

of effects from an environmental perturbation to cortisol to sex hormones to reproductive output in the wild[45]. GC-derived reductions in reproductive function correlate with the decline in egg production witnessed in anemonefish from bleached anemones via a lowering of steroid levels. Even though recruitment of anemonefish is not limited by the bleaching status of anemones[29], GC-derived reductions in anemonefish egg production would still translate into decreased recruitment as previously documented[29] and if prolonged would impact population persistence.

We predict that other species and taxa associated with sea anemones and corals will respond similarly to bleaching events, translating into significant losses in reproductive output. At least 51 species of fishes are facultative symbionts of sea anemones worldwide[46]. Furthermore, scaling up the potential impacts of elevated temperatures on symbiotic fishes in French Polynesia, as many as 12% (56/464) of coastal fish species depend directly, either for food or shelter, on organisms capable of bleaching (Supplementary Data 1). While the strength of such dependency varies greatly, if these species suffer even a fraction of the impact

found for anemonefish, then a short-lived bleaching event could decrease the reproductive output of at least 12% of species, especially those highly dependent on corals or anemones. The cascading effects at the community and ecosystem levels may be considerable, thus hormonal stress responses may have played a role in the population impacts of habitat degradation previously described elsewhere[47].

While no effects on adult anemonefish survival were observed, the effects of bleaching on reproduction and population demography were likely even greater than demonstrated here. The reduction of viable pre-hatching eggs translates into reduced juvenile recruitment[29]. Furthermore, an early exposure to maternal GC levels during embryonic development causes permanent modification of the offspring's HPA/I axis, and may impact future responses to environmental stressors, which may or may not be adaptive, and also may cause trade-offs between life-history traits later in life[48, 49]. The duration of the environmental perturbation will also play an important role on the extent of the demographic consequences of physiological stress. Although bleaching, and their impacts on population demography, was short-lived in Moorea, longer-term climate-induced bleaching events, such as those that occurred on the Great Barrier Reef in 2016 and 2017, could impact fish reproduction over much longer-time scales and cascade into effects at the population level. We expect that demographic effects of physiological stress comparable to those shown here will be found in other systems and taxa exposed to environmental perturbations.

Since anthropogenic stressors and the rate of change in environmental conditions are expected to multiply in the coming decades[50], with bleaching and habitat degradation becoming more frequent, understanding whether individuals and populations can adjust their physiology and behavior fast enough, either plastically or through evolutionary change, is a priority in conservation physiology. Coral reef fish show developmental and transgenerational acclimation in both metabolic rate and reproduction to elevated temperatures[51, 52]. However, in terms of hormonal mechanisms, it is currently unknown if phenotypic flexibility during development that leads to irreversible or reversible modifications of the GC stress response is sufficient to cope with current and predicted future environmental perturbations that exceed the conditions under which this flexibility has evolved[8]. In the few studies that have examined this question, repeated thermal stress suppresses future endocrine sensitivity to acute stressors, for example, in the cane toad[53]. Prenatal maternal physiological stress effects on offspring phenotypes may be another mechanism by which individuals can adjust to climate change[54] (but see ref. [55]). Finally, as the GC stress response is heritable, selection may also act on this mechanism to optimize an individual's fitness[56, 57]; however, such selection has concomitant effects on other behavioral traits and our understanding of how other life-history traits co-vary with the GC stress response is limited. Interestingly, some populations spanning fish to mammals are able to resist environmental and social stresses and breed successfully[4]. Future work determining the mechanisms underlying the resistance of the HPA and/or gonadal axis to stress are crucial to understand if organisms can cope with climate change. In conclusion, the results of this study will be compared with the hormonal stress responses of the same individuals to a second climate-induced bleaching event forecast in 2017 and 2018 to determine the presence of phenotypic flexibility in individual GC responses. As climate change continues to unfold, it is increasingly important to understand how individual differences in GC responses to environmental stressors influence their overall fitness and resilience to further challenges, as well as population vulnerability and persistence.

## Methods

**Study species.** We studied reproduction in free-living orange-fin anemonefish (*A. chrysopterus*) pairs at 13 anemone clusters within the Northern lagoon of Moorea, French Polynesia from October 2015 through November 2016. The 13 anemone clusters all lie within a radius of 2.5 km centered at 17°29′24.86″ S 149°51′29.87″ W (Supplementary Fig. 1). *A. chrysopterus* live mainly in breeding pairs and have an obligate mutualism with their host anemones (*H. magnifica*). *A. chrysopterus* may spawn up to twice per month, but the majority of pairs spawn once around the full moon, with only some individuals spawning for a second time around the new moon. Spawning occurs all year round. Briefly, the male cleans an area of rock under the shelter of the anemone 24 h prior to spawning, and spawning takes place in the late afternoon. Egg clutch size varied from 500 to 3000 eggs. Male anemonefish guard and oxygenate the egg clutch during incubation, which hatches just after sunset on the 6th or 7th night (depending on season).

We also know the locations of an additional 39 anemonefish pairs within the Northern lagoon of Moorea, but that lie within a larger radius of 10 km. Due to their greater distance, these pairs were not monitored for spawning every 2 days, but were bled for hormonal analyses (see below).

**Monitoring reproduction.** All 13 anemone clusters were visited every 2 days over 14 months (October 2015–November 2016) and after assessing behavior that may indicate the presence of an egg clutch, the entire area under the anemone was searched for the presence of eggs. On finding an egg clutch, photos were taken using a SONY camera (DSC-RX100M3) with an underwater Ikelite housing. Due to a change in color and development of eggs we were able to accurately assess the age of the egg clutch (Supplementary Fig. 3) and eggs were usually found within 24 h (day 1) or within 48 h (day 2) after laying. Additional photographs were then taken every 2 days until hatching; therefore we have photographs from each clutch either on days 1, 3, 5, and 7 or on days 2, 4, and 6. The photographs were analyzed using ImageJ and every fertilized egg was counted separately per clutch. We were thus able to measure for each anemonefish pair the following reproductive parameters: monthly spawning frequency, total number of fertilized eggs laid (Day 1/2), mortality over time (Δ Day 6/7 and Day 1/2), and total number of viable pre-hatching eggs at hatching (Day 6/7).

Photographs of all 13 anemones were taken in January 2016. From March 2016 onward we became especially vigilant for anemone host bleaching and photographs of host anemones were taken every week from March until May, after which photographic campaigns occurred on a monthly basis. Photographs before and during peak bleaching are given in Fig. 2.

**Water temperature data.** A continuous time series of water temperature data, measured using bottom-mounted thermistors (Onset HOBOs) at 1 and 6 m on the fringing reef of Moorea, was provided by the Moorea Coral Reef Long-Term Ecological Research site. Temperature data were processed and resampled to a 20 min time step with an accuracy of ±0.2 °C. The average monthly temperature was calculated for the most recent 14-month period (September 2015–October 2016), which includes the warming and bleaching event and average monthly temperatures were calculated for the previous 8 years data (September 2005–August 2015). These data are given in Fig. 1a.

**Blood sampling.** Two scuba divers using a barrier and hand nets captured fish underwater. Blood samples of approximately 0.1 ml per fish were collected out of water, on the boat, laterally from the caudal vein using heparinized 1 ml syringes and kept on ice until processing. Individual blood samples were centrifuged (Sigma Centrifuge 1–14; http://www.sigma-zentrifugen.de) at $10,000 \times g$ for 5 min. The supernatant, a yellow plasma layer, was collected without disturbing the white buffy layer or the blood cells. Stress-induced cortisol response times vary among fishes, from as short as 2.5 min in striped bass *Morone saxatilis*[58] to as long as 120 min in the sea raven *Hemitripterus americanus*[59]. Therefore, the time lapsed from first approaching the anemonefish until blood was flowing in the syringe was kept as short as possible in order to record baseline plasma levels of cortisol (time mean ± SE: females: unbleached = 5.87 ± 1.73 min, bleached = 6.55 ± 1.54 min; males: unbleached = 5.71 ± 1.14 min, bleached = 6.79 ± 1.24 min). Time to capture did not differ significantly between bleached and unbleached anemones (LM: females: $F_{1,22} = 0.085$, $P = 0.774$; males: $F_{1,21} = 0.405$, $P = 0.531$). Furthermore, cortisol did not show a significant regression with time to capture (females: $F_{1,42} = 0.288$, $P = 0.594$, $R^2 = 0.007$, $y = 29.97 - 0.297x$; males: $F_{1,49} = 0.631$, $P = 0.481$, $R^2 = 0.013$, $y = 26.42 + 0.44x$).

Blood was sampled from anemonefish pairs during three periods: April–May 2014 ($n = 20$ pairs); February–March 2016, prior to the warming and bleaching event ($n = 11$ pairs); and between the 3rd and 17th of May 2016, during the peak anemone bleaching event from anemonefish pairs exposed to either bleached ($n = 14$ pairs) or unbleached anemones ($n = 14$ pairs). Cortisol and 11-KT EIA kits have previously been validated for this species[60] and the methods for their measurement are described below, but here we validate the use of the 17β-estradiol EIA kit for these species.

**Endocrinological analyses.** Plasma cortisol, 11-KT, and 17β-estradiol were measured using EIA kits (Cortisol EIA Kit, No. 500360, 11-KT EIA Kit, No.

582751, 17β-estradiol EIA Kit, No. 582251, Cayman Chemicals, SPI BIO, France) as described in ref. [60]. 50 µl of the 7 standards or 50 µl of the serially diluted pooled blood plasma were added with 50 µl of cortisol-, 11-KT-, or estradiol-acetylcholinesterase (AChE) conjugate, respectively, and 50 µl of cortisol-, 11-KT-, or estradiol-specific rabbit antiserum, respectively, to a 96-well plate. Cortisol-, 11-KT-, or estradiol-AChE, and sample or standard cortisol, 11-KT-, or estradiol, competed 4°C for 18, 18 or 1 h respectively, for a limited number of cortisol-, 11-KT-, or estradiol-specific rabbit antiserum binding sites, respectively, whose complex attached to the mouse monoclonal anti-rabbit IgG antibody previously attached to the well. The plate was washed five times to remove any unbound reagents and 200 µl of Ellman's reagent, that contains the substrate to AChE, was added to the wells. The plate was placed on an orbital shaker in the dark for 80 mins. The intensity of the yellow color was measured spectrophotometrically (Beckman Coulter AD 340 Spectrophotometer) at 405 nm and is proportional to the amount of cortisol-, 11-KT-, or estradiol-AChE, respectively, bound to the well, which is inversely proportional to the amount of free cortisol, 11-KT, or estradiol present, respectively. Sample cortisol, 11-KT, or estradiol concentrations, respectively, were determined by interpolation from the standards calibration curve using a common functional model for calibration curves. The data were plotted as % maximum bound (% B/Bo) vs. log concentration using a logit-log curve fit as recommended for this kits (www.caymanchem.com/analysis/eia).

**Kit validation for 17β-estradiol**. Validation of the 17β-estradiol EIA kit comprised: (1) parallel displacement of serially diluted plasma to the standard curve; and (2) precision from intra-assay variability.

Parallelism was evaluated by measuring 17β-estradiol concentrations in pooled plasma samples, serially diluted in EIA buffer provided with the kits. One set of dilution ratios was prepared: 1:3, 1:6, 1:10, and 1:30. The maximum bound (% B Bo$^{-1}$) for each of the four sample dilutions and for the seven standards were plotted against their relative log dilution and the shapes of the resulting curves were compared. These curves must be parallel to support the assumption that the antibody-binding characteristics of standard and sample are similar enough to allow the determination of antibody levels in the diluted plasma sample. An ANCOVA was carried out to determine the homogeneity of slopes between the sample dilutions and those of the kit's standards. In addition, regression analysis of the diluted sample was used to determine the dilution factor that corresponds to 50% of antibody bound. Four dilutions of pooled plasma from *A. chrysopterus* ran linear and were thus were screened with the 17β-estradiol kit's standard curve. The two lines were found to run parallel (Supplementary Table 10 and Supplementary Fig. 4). Regression analyses enabled the appropriate dilution factors for 50% of antibody bound for the anemonefish to be determined with the 17β-estradiol kit (Supplementary Table 10).

Precision was assessed by examining intra-assay variability of samples with different hormones levels. Intra-assay variability was determined by evaluating three plasma samples in duplicate within the same run of the assay. The variability or coefficients of variation (CV) of repeated measures of samples were assessed. CV was calculated according to the formula: $CV = (SD \ \overline{X}^{-1}) \times 100$. A kit was considered to have good precision if the CV was <20% as per the guidelines (see in ref. [60]). *A. chrysopterus* also showed high accuracy and precision with the 17β-estradiol kit determined from intra-assay variability; 7.3% ($n = 3$).

In conclusion, the dose-response curves were parallel to the 17β-estradiol EIA assay kit standards (Supplementary Fig. 4; Supplementary Table 10) and high precision was obtained from intra-assay variability (<10%) with *A. chrysopterus*. Consequently, this kit can be confidently used for measuring 17β-estradiol in *A. chrysopterus*.

**Data availability**. The endocrinological and reproductive data that support the findings of this study are available in the Dryad Data Repository with the identifier (doi:10.5061/dryad.0k24s).

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

## Acknowledgements

Financial support was provided by the Agence Nationale de la Recherche (ANR-11-JSV7-012-01/Live and Let Die) to S.C.M. and (ANR-14-CE02-0005-01/Stay or Go) to Glenn Almany, S.C.M. and R.B., LabEx "CORAIL" ("Where do we go now?" to R.B. and S.C.M.), Fundação para a Ciência e Tecnologia (SFRH/BPD/26901/2006) to R.B. and to the European Commission's Erasmus program that made possible for RO to carry out a traineeship at CRIOBE as part of an agreement between Ulster University (Northern Ireland) and the EPHE. We thank Gerrit Nanninga, Georgia McDowell, Oscar Prado Merini, Adri Sparks, Lauriane Derrien, Tara Cousins, and Zoe Scholz for assistance with data collection; Chris Gotschalk, Gastil Buhl, and Andy Brooks from the LTER for temperature data; Nathalie Tolou for logistical help; David Duneau for graphical help; and Isabelle Côté for constructive comments on the manuscript.

## Author contributions

Designed the study: R.B., S.C.M. Collected the data: R.B., A.B., R.O.D., S.C.M. Analyzed the data: R.B., S.C.M. Wrote and revised the manuscript: all authors.

## Additional information

**Competing interests:** The authors declare no competing financial interests.

