## [Peer Review File · Nature Communications]

Reviewers' comments:

Reviewer #1 (Remarks to the Author):

This interesting paper examines the effects of anemone bleaching, caused by anomalously high water temperature, on the hormonal stress response in anemonefishes and how this is associated with a serious decline in reproduction. The study is novel and significant in: 1) tracking anemones and their host fishes through a natural thermal anomaly in the wild and, 2) tracking both bleached and unbleached anemones over the same time period to demonstrate the effects on the fish are likely due to bleaching of their host anemone, not the direct effects of higher temperature. Overall, the results are compelling and demonstrate how stress caused to the biogenic habitat (anemones) by warming can have substantial effects on fishes that depend on these habitats, through a cascading effect of stress on reproductive hormones, ultimately leading to decreased reproduction. I have just a number of relatively minor comments.

1. At line 129 you emphatically state that all anemonefish experienced the same thermal stress, and thus the effects must have been caused by the host bleaching and not heterogeneity of thermal effects directly on the fish. Did you have individual temperature loggers to demonstrate this? For the most part, the bleached and non-bleached anemones are not co-located – importantly, there is only one bleached anemone (#6) that is immediately co-located with non-bleached anemones. Can you further rule out within-reef variation in the pattern of warming that the anemones (and thus the fish) experienced?

2. For the hormone concentrations shown in Figure 2 you sampled anemonefish from more than just the 13 anemones shown in Figure 1 and you sampled these individuals on two occasions, before and during the bleaching event. It would be sensible to include individual fish (e.g. anemonefish ID) as a random effect in the linear mixed effects model (LMM) to help account for individual variation in hormone levels, which may vary greatly as evident in Figure 2. This random effect may also help account for the uneven spatial arrangement of bleached and non-bleached anemones mentioned above.

3. Line 138. It is not clear what you mean by “Differences in baseline levels of GCs are considered the most appropriate measure for assessing the impacts of enduring, long-term environmental challenges...” What is the baseline you are referring to and what are you comparing? Moreover, this statement needs some supporting evidence and citations. I agree that GC levels may provide good information about short-term stress responses to environmental change, but is there also good support for a GCs as an indicator of long-term (i.e. chronic) environmental change. I would have thought that GCs are more likely to return towards levels of unstressed individuals over the longer term, as they acclimate to the environmental change.

4. Line 148. I think it's overstated to say that “GC-derived reductions in reproductive function are “clearly responsible” for the decline in egg production witnessed in anemonefish from bleached anemones via a lowering of steroid levels.” You have a very clear association between GC levels and reproduction, but you haven't independently manipulated GC levels in a way that would demonstrate cause and effect. Just a bit more caution needed in the wording here.

5. Line 185-191. This important section has missed key research that examines developmental plasticity and maternal effects on the response to warming in reef fishes, including effects on reproduction. For example:

Donelson et al. (2014). Reproductive acclimation to increased water temperature in a tropical reef

fish. PLoS One 9 e97223. doi:10.1371/journal.pone.0097223.

Donelson et al. (2016). Transgenerational plasticity of reproduction depends on rate of warming across generations. *Evolutionary Applications*, doi: 10.1111/eva.12386.

These papers deal directly with the topic being discussed here and are more relevant to the current study than the cited papers on cane toads and primates! They show that reproductive capacity at +1.5C in a reef fish is restored to current-day control temperatures when the fish develop at the warmer temperature and that gradual warming over two generations induces even greater reproductive plasticity.

6. Figure 2. The legend needs to explain why there are many more samples for each period than just the N=7 blue (non-bleached) and N=6 red (bleached) anemones shown in Figure 1.

Line 387. It was 13 anemones at several location, not 13 locations.

7. Line 403 & 417. 13 anemones, not sites.

8. Line 422. What is the physical location of the bottom-mounted thermistors relative to the 13 anemones surveyed here?

9. Line 436. How often was it not possible to get blood samples within 3 minutes? Also, was the time from capture to sampling equal for fish from bleached and non-bleached anemones? This is important because even 3 minutes would be sufficient time post-capture to potentially observe stress related changes in cortisol levels.

10. Line 430. More information is needed here on how the anemonefish were collected, handled and processed. This is critical for understanding any possible effects of the capture and handling process on cortisol levels. Also, was blood taken underwater or were the fish transferred to a boat for this process. This important information is missing.

Reviewer #2 (Remarks to the Author):

Overall I enjoyed reading this manuscript. It is well-written and easy to read. The physiological links between climate change and demographic changes are mostly unknown, although there have been a number of suggestions in the literature that the hormonal stress system plays a role. The connection between bleaching and fish reproduction that is detailed here is a unique data set that will be of great interest to many researchers. It is a great example of the burgeoning field of conservation physiology. I only had a few, mostly minor, comments.

Comments:

1. Line 32: I would suggest that concluding that this is a causal link should be tempered. There is certainly a strong correlation, but bleaching might not be the ultimate cause of the increase in cortisol, as suggested by the authors themselves in lines 131-135 (e.g. increase in perceived risk of predation, decreased anemone toxicity, etc.). Furthermore, the concomitant decrease in sex steroids and fecundity are not experimental, so claiming a causal link is again premature. There are other reasons sex steroids and fecundity can decrease irrespective of cortisol (e.g. reduction in appropriate mating signals – so that the use of “clearly” in Line 148 is perhaps not so clear). In fact, contrary to the claim in Lines 142-144, sex steroid regulation by GCs is thought to occur mostly at stress-induced, not

baseline, concentrations of GCs. In sum, although I think that these are important and strong correlations among these variables, it is premature to make the many claims in this manuscript that the relationships are causal.

2. Lines 52-53: The statement that “examples of stress responses, and their regulatory impacts, to climate change in wild animals are lacking” is not accurate. There have been a number of examples of this. The following are two examples that come to mind, but this is not an exhaustive list.

a. Impact of a warming ocean on kittiwake survival and reproduction:

Buck, C.L., O'Reilly, K.A., Kildaw, S.D., 2007. Interannual variability of Black-legged Kittiwake productivity is reflected in baseline plasma corticosterone. *Gen Comp Endocrinol* 150, 430-436.

Kitaysky, A.S., Piatt, J.F., Wingfield, J.C., 2007. Stress hormones link food availability and population processes in seabirds. *Mar. Ecol.-Prog. Ser.* 352, 245-258.

b. Impact of El Niño on Galapagos marine iguana survival:

Romero, L.M., Wikelski, M., 2001. Corticosterone levels predict survival probabilities of Galápagos marine iguanas during El Niño events. *Proc Natl Acad Sci, USA* 98, 7366-7370.

Romero, L.M., Wikelski, M., 2010. Stress physiology as a predictor of survival in Galapagos marine iguanas. *Proc R Soc Lond B* 277, 3157-3162.

3. Lines 61-64: See again the references above.

4. Lines 93-94 and Figures 2A and B: The text indicates that the three sampling periods reflect before, during, and after the bleaching event, whereas the figures depict two periods before and one period during bleaching. Although other data include the post bleaching period (e.g. Fig. 3), it would be useful to indicate why hormones were not measured from fish in the post-bleaching period as well.

5. Line 142: A recent review, however, indicated that there is actually little support from the literature that elevated baseline cortisol is a common response to long-term environmental perturbations.

Dickens, M.J., Romero, L.M., 2013. A consensus endocrine profile for chronically stressed wild animals does not exist. *Gen Comp Endocrinol* 191, 177-189.

Minor Comments:

1. There are a number of typographical errors in the reference list that need to be fixed.

2. Lines 87-88: Unless I'm missing something, only one panel for Fig. S1 is provided, so distinguishing between S1A and S1B is not necessary.

It takes three to tango: Cascading fitness effects of anemone bleaching on associated anemonefish hormonal stress-response, reproductive hormones and reproduction

Replies to reviewers' comments:

Reviewer #1:

This interesting paper examines the effects of anemone bleaching, caused by anomalously high water temperature, on the hormonal stress response in anemonefishes and how this is associated with a serious decline in reproduction. The study is novel and significant in: 1) tracking anemones and their host fishes through a natural thermal anomaly in the wild and, 2) tracking both bleached and unbleached anemones over the same time period to demonstrate the effects on the fish are likely due to bleaching of their host anemone, not the direct effects of higher temperature. Overall, the results are compelling and demonstrate how stress caused to the biogenic habitat (anemones) by warming can have substantial effects on fishes that depend on these habitats, through a cascading effect of stress on reproductive hormones, ultimately leading to decreased reproduction. I have just a number of relatively minor comments.

1. At line 129 you emphatically state that all anemonefish experienced the same thermal stress, and thus the effects must have been caused by the host bleaching and not heterogeneity of thermal effects directly on the fish. Did you have individual temperature loggers to demonstrate this? For the most part, the bleached and non-bleached anemones are not co-located – importantly, there is only one bleached anemone (#6) that is immediately co-located with non-bleached anemones. Can you further rule out within-reef variation in the pattern of warming that the anemones (and thus the fish) experienced?

Firstly, unfortunately no, we did not have individual temperature loggers at each of the sites in 2016.

Secondly, we have only added the locations of the anemones with breeding anemonefish pairs to Figure S1 and while there is only one bleached anemone (6) that is co-located with non-bleached anemones in Fig S1, in reality there were and are a lot more co-located bleached and un-bleached anemones but that do not have breeding anemonefish pairs so they were not included in this study.

Thirdly, to rule out within-reef variation in the pattern of warming that the anemones experienced (i.e. temperature) we do have supplementary data. During the continued bleaching event this year in 2017, which we are currently monitoring, we placed temperature meters at 9 anemones, some at the same anemones as last year (7 out of the 13 anemones, 4 bleached and 3 unbleached; plus at an extra unbleached anemone and at an extra bleached anemone) between 21st march and 11th may 2017. The anemones that bleached in 2016 also bleached in 2017 and vice versa for unbleached anemones. In order to rule out within-reef variation in the pattern of warming for 2017 we have tested for a difference in max and average daily temperatures recorded by the data loggers between anemones that bleached or not. 1) we did not find a significant difference in temperatures between bleached and unbleached sites (see Table S7 and Fig

S2). However, this data is for the 2017 bleaching event. Therefore, to validate, as best we can, these short-term data loggers in 2017 with the long-term temperatures monitoring carried out by the LTER that we report, we also compared 2017 temperatures with those recorded in 2016 for the same period (21st march and 11th may 2016). We did not find a significant difference in temperatures between either bleached and unbleached anemones in 2017 and the temperatures recorded for the same period in 2016 (see Table S7 and Fig S2). Therefore, we believe we can safely rule out within-reef variation in the pattern of warming at our sites for 2017 and we can infer the same for 2016.

We have now added this to the paper and referenced the Figure and Table in the supplementary material and thus we have the sentence unchanged.

2. For the hormone concentrations shown in Figure 2 you sampled anemonefish from more than just the 13 anemones shown in Figure 1 and you sampled these individuals on two occasions, before and during the bleaching event. It would be sensible to include individual fish (e.g. anemonefish ID) as a random effect in the linear mixed effects model (LMM) to help account for individual variation in hormone levels, which may vary greatly as evident in Figure 2. This random effect may also help account for the uneven spatial arrangement of bleached and non-bleached anemones mentioned above.

Anemonefish ID was already included in the model as a random effect, as the results showed that a random effect was not needed for the majority of tests (except the tests that were non-significant i.e. female 11KT and male estradiol) we did not think it necessary to report the results, despite including the random effect in the model. However, now we have added these results to the supplementary material in Table S1B.

3. Line 138. It is not clear what you mean by “Differences in baseline levels of GCs are considered the most appropriate measure for assessing the impacts of enduring, long-term environmental challenges...” What is the baseline you are referring to and what are you comparing? Moreover, this statement needs some supporting evidence and citations. I agree that GC levels may provide good information about short-term stress responses to environmental change, but is there also good support for a GCs as an indicator of long-term (i.e. chronic) environmental change. I would have thought that GCs are more likely to return towards levels of unstressed individuals over the longer term, as they acclimate to the environmental change.

Briefly, a baseline GC level refers to the level of cortisol in an unstressed individual, i.e. the cortisol levels that a fish experiences as a daily routine.

In more detail, the stress response can be split into an acute response (which is considered temporary – minutes to hours) whereby the hypothalamic-pituitary-adrenal (HPA) axis is initiated and culminates in the secretion of glucocorticoid (GC) hormones such as cortisol. Such levels of cortisol are referred to as stress-induced cortisol levels and can be compared to baseline cortisol levels, which are levels before the HPA axis has been triggered by a stressor. Once the stressor has passed, cortisol levels return to baseline level via negative feedback of the HPA axis.

In contrast, a chronic stress response can be caused by long-term release of GCs (days or

weeks) that can disrupt the reproductive hormone axis and reproductive behavior. A review in 2009 by Bonier et al found that overall, the majority of the available data supports a positive relationship between baseline cortisol levels and environmental challenges i.e. high baseline cortisol levels over long periods of time indicate that the individual is chronically stressed. Indeed as you suggest habituation to a continuous stressor can occur, baseline levels should return to pre-exposure levels and thus chronic stress should not be observed. However, in our paper, 3 months after the bleaching event, baseline cortisol is still elevated, suggesting that these levels have NOT returned to unstressed levels and thus represent a chronic stress.

In terms of the best endocrine measure to use to measure chronic stress, thanks to reviewer 2, I also now cite a more recent paper (Dickens et al 2014), which finds that there is currently no consensus as to the best endocrine response for chronic stress, and thus we have modified this part of the text.

4. Line 148. I think it's overstated to say that "GC-derived reductions in reproductive function are "clearly responsible" for the decline in egg production witnessed in anemonefish from bleached anemones via a lowering of steroid levels." You have a very clear association between GC levels and reproduction, but you haven't independently manipulated GC levels in a way that would demonstrate cause and effect. Just a bit more caution needed in the wording here.

As suggested, we have downplayed this part of our results and no longer refer to a causal link, rather just a correlation.

5. Line 185-191. This important section has missed key research that examines developmental plasticity and maternal effects on the response to warming in reef fishes, including effects on reproduction. For example:

Donelson et al. (2014). Reproductive acclimation to increased water temperature in a tropical reef fish. PLoS One 9 e97223. doi:10.1371/journal.pone.0097223.

Donelson et al. (2016). Transgenerational plasticity of reproduction depends on rate of warming across generations. Evolutionary Applications, doi:10.1111/eva.12386.

These papers deal directly with the topic being discussed here and are more relevant to the current study than the cited papers on cane toads and primates! They show that reproductive capacity at +1.5C in a reef fish is restored to current-day control temperatures when the fish develop at the warmer temperature and that gradual warming over two generations induces even greater reproductive plasticity.

We have added a discussion of these important papers to the discussion. However, we have decided to leave in the references on cane toads and primates as they specifically discuss acclimation of hormonal mechanisms.

6. Figure 2. The legend needs to explain why there are many more samples for each period than just the N=7 blue (non-bleached) and N=6 red (bleached) anemones shown in

Figure 1.

Hormonal measures were taken from an additional 39 anemonefish pairs that were not monitored for spawning every two days because they required a greater distance to travel to them which was not feasible every two days for 14 months! Sample sizes and an explanation have been added to the Figure legend (now Fig 3).

Line 387. It was 13 anemones at several location, not 13 locations.

Changed to anemone clusters (as most "sites" have more than one anemone).

7. Line 403 & 417. 13 anemones, not sites.

Changed to anemone clusters (as most "sites" have more than one anemone).

8. Line 422. What is the physical location of the bottom-mounted thermistors relative to the 13 anemones surveyed here?

We have now added the physical location of the bottom-mounted thermistors to Fig. S1 and due to the analysis described above, which is now in the supplementary material, we assume that the physical location is representative of the temperatures that the anemones and fish in our study would have experienced.

9. Line 436. How often was it not possible to get blood samples within 3 minutes? Also, was the time from capture to sampling equal for fish from bleached and non-bleached anemones? This is important because even 3 minutes would be sufficient time post-capture to potentially observe stress related changes in cortisol levels.

We have modified this part in the text to better explain blood sampling. We now provide mean times from first approaching the fish till blood was flowing in the syringe. We also present the results that demonstrate no significant differences in the time to capture between bleached and unbleached anemones in 2016. We also show that there is no significant regression of cortisol level with time to capture.

10. Line 430. More information is needed here on how the anemonefish were collected, handled and processed. This is critical for understanding any possible effects of the capture and handling process on cortisol levels. Also, was blood taken underwater or were the fish transferred to a boat for this process. This important information is missing.

We have modified this part in the text to better explain blood sampling as requested.

Reviewer #2 (Remarks to the Author):

Overall I enjoyed reading this manuscript. It is well-written and easy to read. The physiological links between climate change and demographic changes are mostly unknown, although there have been a number of suggestions in the literature that the hormonal stress system plays a role. The connection between bleaching and fish reproduction that is detailed here is a unique data set that will be of great interest to many researchers. It is a great example of the burgeoning field of conservation physiology. I only had a few, mostly minor, comments.

Comments:

1. Line 32: I would suggest that concluding that this is a causal link should be tempered. There is certainly a strong correlation, but bleaching might not be the ultimate cause of the increase in cortisol, as suggested by the authors themselves in lines 131-135 (e.g. increase in perceived risk of predation, decreased anemone toxicity, etc.). Furthermore, the concomitant decrease in sex steroids and fecundity are not experimental, so claiming a causal link is again premature. There are other reasons sex steroids and fecundity can decrease irrespective of cortisol (e.g. reduction in appropriate mating signals – so that the use of “clearly” in Line 148 is perhaps not so clear). In fact, contrary to the claim in Lines 142-144, sex steroid regulation by GCs is thought to occur mostly at stress-induced, not baseline, concentrations of GCs. In sum, although I think that these are important and strong correlations among these variables, it is premature to make the many claims in this manuscript that the relationships are causal.

We agree with the reviewer and thank the reviewer for the interesting insight. As suggested we have removed mention of a causal link and have tempered this to a correlation both on line 32 and line 148.

2. Lines 52-53: The statement that “examples of stress responses, and their regulatory impacts, to climate change in wild animals are lacking” is not accurate. There have been a number of examples of this. The following are two examples that come to mind, but this is not an exhaustive list.

a. Impact of a warming ocean on kittiwake survival and reproduction:

Buck, C.L., O'Reilly, K.A., Kildaw, S.D., 2007. Interannual variability of Black-legged Kittiwake productivity is reflected in baseline plasma corticosterone. *Gen Comp Endocrinol* 150, 430-436.

Kitaysky, A.S., Piatt, J.F., Wingfield, J.C., 2007. Stress hormones link food availability and population processes in seabirds. *Mar. Ecol.-Prog. Ser.* 352, 245-258.

b. Impact of El Nino on Galapagos marine iguana survival:

Romero, L.M., Wikelski, M., 2001. Corticosterone levels predict survival probabilities of Galápagos marine iguanas during El Niño events. *Proc Natl Acad Sci, USA* 98, 7366-7370.

Romero, L.M., Wikelski, M., 2010. Stress physiology as a predictor of survival in Galapagos marine iguanas. *Proc R Soc Lond B* 277, 3157-3162.

Indeed we had overlooked these papers, they have now been included and we were not familiar with all of them.

3. Lines 61-64: See again the references above.

These references have now been included.

4. Lines 93-94 and Figures 2A and B: The text indicates that the three sampling periods reflect before, during, and after the bleaching event, whereas the figures depict two periods before and one period during bleaching. Although other data include the post bleaching period (e.g. Fig. 3), it would be useful to indicate why hormones were not measured from fish in the post-bleaching period as well.

This point was also raised by reviewer 1 and we have modified the text so that the timing

of the hormonal measures versus the monitoring of spawning is clearer. Hormones were not measured from fish in the post-bleaching period due to lack of time as this post-bleaching period was a very busy field season for other field projects related to and others unrelated to anemonefish. It is unfortunate, but we had other commitments with visiting researchers that took priority.

5. Line 142: A recent review, however, indicated that there is actually little support from the literature that elevated baseline cortisol is a common response to long-term environmental perturbations.

Dickens, M.J., Romero, L.M., 2013. A consensus endocrine profile for chronically stressed wild animals does not exist. *Gen Comp Endocrinol* 191, 177-189.

Thank you for alerting our attention to this paper.

While a review in 2009 by Bonier et al found that overall the majority of the available data supports a positive relationship between baseline cortisol levels and environmental challenges i.e. high baseline cortisol levels over long periods of time indicate that the individual is chronically stressed, clearly this more recent review (Dickens et al 2014), finds that there is currently no consensus as to the best endocrine response for chronic stress. We have now changed the text accordingly and added this new reference.

Minor Comments:

1. There are a number of typographical errors in the reference list that need to be fixed.

We have now checked the reference list and corrected the mistakes that we found.

2. Lines 87-88: Unless I'm missing something, only one panel for Fig. S1 is provided, so distinguishing between S1A and S1B is not necessary.

We were actually referring to Fig. S2, so thank you and this has been corrected.